# Non-Destructive Method for Estimating Seed Weights from Intact Peanut Pods Using Soft X-ray Imaging

Guangjun Qiu [1,2], Yuanyuan Liu [2,3], Ning Wang [2,*], Rebecca S. Bennett [4] and Paul R. Weckler [2]

1. Institute of Facility Agriculture of Guangdong Academy of Agricultural Sciences, Guangzhou 510640, China; qiuq16@scau.edu.cn
2. Department of Biosystems and Agricultural Engineering, Oklahoma State University, Stillwater, OK 75078, USA; liuyuanyuan1980@126.com (Y.L.); paul.weckler@okstate.edu (P.R.W.)
3. College of Information Technology, Jilin Agricultural University, Changchun 130118, China
4. U.S. Department of Agriculture-Agricultural Research Service, Peanut and Small Grains Research Unit, Stillwater, OK 74075, USA; rebecca.bennett@ars.usda.gov
* Correspondence: ning.wang@okstate.edu; Tel.: +1-405-744-2877

**Abstract:** In the U.S., peanut farmers receive premium prices for crops with high seed grades. One component of seed grade is the proportion of seed weight to that of pod hulls and other matter. Seed weight and size are also important traits for food processors. Current methods for evaluating peanut seed grade require the opening of the pod and are time-consuming and labor-intensive. In this study, a non-destructive and efficient method to determine peanut seed weights was investigated. X-ray images of a total of 513 peanut pods from three commercial cultivars, each representing three market types, were taken using a soft X-ray imaging system. The region of interest of each image, the seeds, was extracted two ways, manually and with a differential evolution segmentation algorithm. The comprehensive attenuation index (CAI) value was calculated from the segmented regions of interest. Lastly, linear regression models were established between peanut seed weights and the CAI. The results demonstrated that the X-ray imaging technology, coupled with the differential evolution segmentation algorithm, may be used to estimate seed weights efficiently from intact peanut pods.

**Keywords:** image processing; differential evolution; weight estimation; image segmentation; automation



## 1. Introduction

Peanut (*Arachis hypogaea* L.), an affordable and nutritionally dense source of protein, fatty acids, and carbohydrates [1], is widely grown in tropical and subtropical regions around the world [2,3]. In 2022, approximately 50 million metric tons were harvested worldwide, with China being the largest producer, with 37% of the world's total production [4].

In the U.S., official grading procedures are used to determine prices to peanut growers for their harvested, unshelled crops or "farmers stock peanuts" [5,6]. A critical component determining the value of farmers' stock peanuts is the percentage of total sound mature kernels (TSMK), i.e., undamaged kernels of a specific minimum size from a sample including pod hulls, foreign material, and small and damaged kernel [5,6]. To determine the TSMK for runner, Spanish, and Virginia peanut market types, 500 g pod samples must be sorted by size using pre-sizing machines before being machine shelled. After shelling, the seeds are then sorted by size using mechanical shakers and visually inspected for damage. Throughout the process, each category (TSMK, hulls, damaged kernels, etc.) is manually weighed [5,6]. The official process for grading and inspection is time-consuming and labor-intensive, and some service costs are passed on to growers.

Several efforts have been made to make the peanut grading process more efficient. Dowell [7,8] designed an automated system by interfacing a digital balance and a digital

moisture meter for cleaning, shelling, and sizing seeds and pods, and data collection, but the system was not adopted by the industry. Few changes have been made to the official grading standards since its establishment in the 1960s [9]. Based on the observations that pod density is positively correlated with pod maturity and larger seeds [10], Butts and colleagues investigated the ability of using bulk pod density to predict grade factors [9]. They found that TSMK increased as pod bulk density increased; however, the association was not close enough to be useful for seed grading purposes [9]. Another non-destructive method—digital image analysis—was used to indirectly estimate pod volume associated with pod density and seed percentage by weight. Wu and colleagues (2015) found that pod volume was negatively correlated between pod density (r = −0.59) and percent seed (r = −0.34) [11].

Soft X-ray imaging is another non-destructive technique that has the potential to estimate seed weights from intact pods. X-ray images record levels of X-ray attenuation caused by chemical composition, density, and thickness [12] and, thus, can be used to analyze the internal structure of samples. Soft X-rays are electromagnetic waves with a wavelength range of 0.1–10 nm. Because they have low photon energies (about 0.1–5 keV) relative to hard X-rays (about 5–100 keV with a wavelength range of 0.01–0.1 nm), soft X-rays are considered to be safer to operate. At present, soft X-ray imaging technique has been widely used for detecting residual bones in meat such as chicken, pork and fish during processing link by in-line scanning X-ray images of samples in agricultural domain. Soft X-ray imaging has been recommended by the International Seed Testing Association (ISTA) for determining seed quality [13–15]. Furthermore, the attenuation caused by biological tissue such as crop seeds for X-ray radiation is commonly lower than other industrial materials. Hence, soft X-rays are more suitable for obtaining well-contrasted X-ray images than hard X-rays for biological tissue analysis. Previous research has reported innovative results by applying a soft X-ray imaging system for seed quality detection. For example, soybean seeds with physical and insect damage were detected with soft X-ray imaging [16]. Using extracted histogram and textural features from X-ray images and quadratic discriminant analysis, Chelladurai et al. [17] identified hollow spaces excavated by weevils and weevil pupae with high accuracy (≥89%). But eggs and pupae were more difficult to detect due to the detection limits of the system. Soft X-ray imaging technique was also effective on estimating pecan nutmeat weight, and the method resulted in an error of less than 10% compared with the reference method [18]. Recently, artificial intelligence tools and information fusion methods were also reported in analyzing X-ray images for seed quality detection. By analyzing the merger data from FT-NIR spectroscopy and X-ray images of small forage grass seeds, Medeiros et al. [19] could predict seed germination with a high accuracy of 90% by using machine learning methods. Nadimi et al. [20] achieved automated detection of mechanical damage in flaxseeds by utilizing an X-ray imaging technique with machine and deep learning tools, and the classification accuracy was as high as 91.0%. Hong et al. [21] could obtain a high accuracy of 92.51% by establishing an ensemble-based information fusion model from hyperspectral and X-ray image data for predicting the viability of pepper seeds. A more advanced (and costly) application of X-ray technology, X-ray micro-computed tomography (X-ray μCT), can be used to generate three-dimensional images. Guelpa and colleagues [22] used X-ray μCT on maize seeds to determine the volume and density of maize seed for assessing milling quality.

To date, few works have been reported on applying X-ray technologies to evaluate peanut seed quality. Sorenson et al. [23] explored X-ray imaging as an alternative to the commonly-used hull scrape method [24] for assessing peanut maturity. They found that the seed area and density features of dried peanut pods were highly correlated with some stages of maturity [23]. Recently, Domhoefer et al. [25] also focused on investigating the possibilities of estimating peanut seed weight by using the X-ray technique. They obtained good models while applying the convolutional neural network (CNN) model on analyzing X-ray images, and high determination coefficients of 0.94 for kernel weight were obtained by the CNN method in the testing steps, respectively. This study attempted to deduce the

computationally relationship between the gray level value of X-ray image and peanut seed weight, and then to investigate the method of using X-ray imaging techniques to estimate peanut seed weights from intact peanut pods. This was achieved through four specific tasks: (1) establish an X-ray imaging system to collect X-ray images of the peanut pods; (2) develop image processing and segmenting algorithms to extract the peanut seed area in X-ray image as the region of interest (ROI); (3) deduce the mathematical formula between the grey value of pixels of ROI and the peanut seed weight; and (4) conduct tests to validate the developed algorithm and method.

## 2. Materials and Methods

### 2.1. Peanut Samples

A total of 513 pods from three commercial peanut cultivars (Florida Fancy, Plant Variety Patent (PVP) 200800231; Wynne, PVP 201500288; OLé [26]) were used for the experiment. Florida Fancy and Wynne are Virginia market-type peanuts with large pods, and OLé is a Spanish market-type peanut with small pods. Peanut pods without external damage were randomly selected.

### 2.2. Soft X-ray Imaging System

Figure 1 shows a schematic diagram of the soft X-ray imaging system used for this study. The soft X-ray imaging system used in this study was similar to that used by Kotwaliwale et al. [18], upgrading an X-ray camera with a larger imaging unit. It consisted of an X-ray tube (XTFTM-5011, Oxford Instruments, X-ray Technologies, Inc., Scotts Valley, CA, USA), a solid-state digital X-ray camera (Shad-o-BoxTM 2048, Rad-icon, Imaging, Corp., Santa Clara, CA, USA), a digital frame grabber (Imagenation® PXD 1000, Imagenation Corp., Minneapolis, MN, USA), a data acquisition and control card (Omega® DAQ 801 OM, Omega Engineering, Inc., Norwalk, CT, USA), and a personal computer with ShadoCam software Version 3.0.2 (Rad-icon Imaging, Corp., Santa Clara, CA, USA).

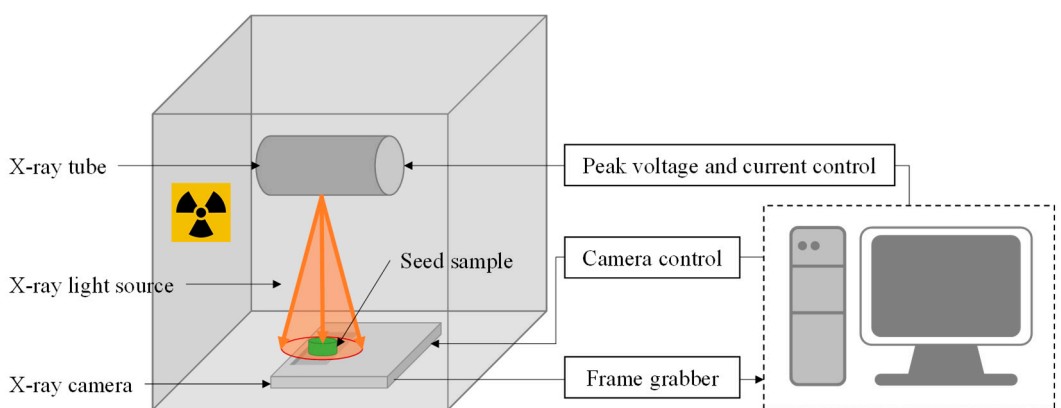

**Figure 1.** The schematic diagram of the soft X-ray imaging system.

The X-ray tube operated at a voltage range from 0 to 50 kVp with a maximum current of 1 mA, which has a lower peak voltage but higher maximum current than those parameters in other X-ray imaging systems [25]. Increasing the acceleration peak voltage and/or current might be a feasible way to shorten the integration time for capturing the X-ray images. The energy of X-ray beams generated by these peak voltage and current ranges is suitable for agricultural products [27]. The voltage and current of the X-ray tube in this study could be adjusted in 5 kVp and 0.1 mA increments, respectively. Inside the X-ray tube, the X-ray beam was generated at the tungsten anode and passed through a beryllium window with a thickness of 127 μm. The X-ray tube emitted a cone-shaped beam with an angle of 25 degrees at a mounting height of 18.0 cm above the X-ray camera.

The X-ray camera contained a 49.2 mm × 98.6 mm area image sensor, which had a 1024 pixels × 2048 pixels photodiode array with 48 μm pixel spacing. With a $Gd_2O_2S$

scintillator screen in contact with the photodiode array, the incident X-ray photons were converted to visible light signals. The analog signals in each pixel were then digitized to binary codes and processed by the frame grabber, which was installed in a desktop computer with an Intel i5-8250U CPU and a 16 GB RAM, running Microsoft Windows 10 Pro. X-ray images of peanut samples were captured and saved as gray-level images.

### 2.3. Imaging Peanut Pods

During every image acquisition, radiograph images of two peanut pods were taken by placing the pods on the imaging window of the X-ray camera. Since the mass attenuation coefficients vary with the material of the samples and the energy of the X-ray light source, two assumptions were made [22]. First, this study assumed that the mass attenuation coefficients of peanut seeds in a single cultivar as well as in three cultivars were uniform. Second, the emitted energy of the X-ray light remained the same during an imaging acquisition when the X-ray tube was set at the same peak voltage (Vp) and current. During the X-ray image acquisition, the X-ray source was configured at 20 kVp voltage, 0.6 mA current, and 2 s integration time, which were optimized in pre-experiment according to the method of literature [18].

### 2.4. Manual Segmentation and Comprehensive Attenuation Index

All X-ray images were preprocessed with noise filtering, cropping, and compression conversion. Due to some defective pixels inside the X-ray camera detector, a median filter with a window size of $5 \times 5$ pixels was first applied to remove image noise. It was also found that pixels near the border of all collected X-ray images were either inactive or responded erratically. Therefore, each X-ray image of single pods was manually cropped to $1000 \times 512$ pixels.

Figure 2 shows the segmentation and calculation processes for a peanut X-ray image. Each image was segmented manually with selected intensity thresholds to obtain the shell and seed regions. For example, in Figure 2a, it was observed that attenuation levels were different when an X-ray beam passed through peanut shells (brighter) and seeds (darker). The X-ray beam was attenuated more when passing through the seed (darker region) than the shell (brighter region). By using the selected thresholds, the region of seed was extracted as the region of interest (ROI) in Figure 2c.

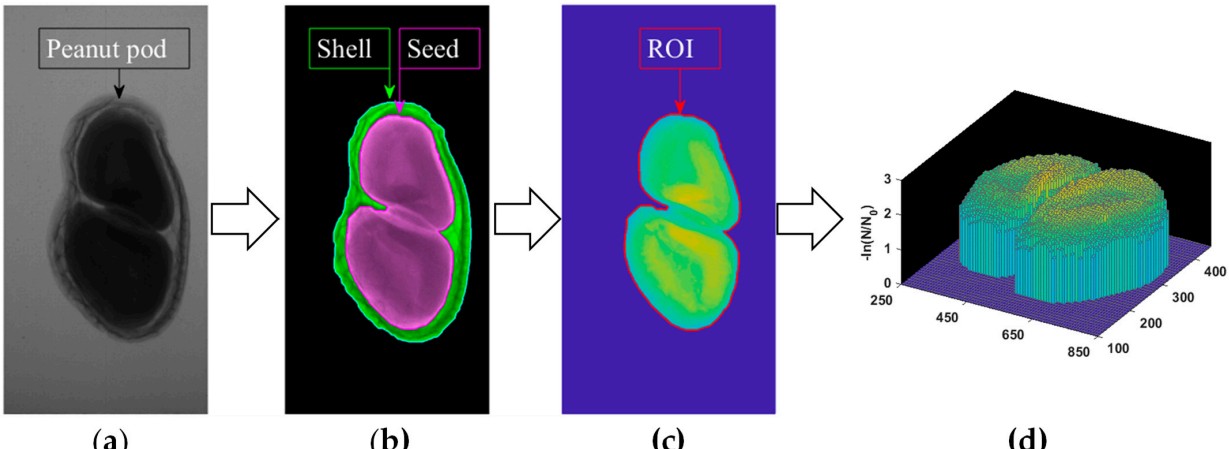

**Figure 2.** The segmentation and calculation process for a peanut X-ray image: (**a**) a raw X-ray image of a peanut pod. (**b**) Identification of the shell region in green and the seed region in purple. (**c**) Extracted seed region and marked as ROI. (**d**) Virtual volume at each pixel of ROI according to comprehensive attenuation index (CAI) value.

Calculating the total amount of attenuation within an ROI was essential to establishing a weight calculation model. Linear attenuation and mass attenuation coefficients were

used to calculate attenuation in monochromatic radiation, and both were specific to the targeted object and the energy level of the X-ray beam [28]. The mathematic formula used to describe the attenuation is shown in Equation (1) [28]:

$$N = N_0 \cdot e^{-\mu x} \tag{1}$$

where $N$ is the number of transmitted photons, $N_0$ is the number of incident photons, $e$ was the base of the natural logarithm, $\mu$ is the linear attenuation coefficient $(m^{-1})$, and $x$ is the thickness of a target object (m). The mass attenuation coefficient is often used in diagnostic radiology and is defined as the ratio of the linear attenuation coefficient and matter density [28]:

$$\gamma = \frac{\mu}{\rho} \tag{2}$$

where $\gamma$ is the mass attenuation coefficient $(m^2/kg)$, $\rho$ is the physical density of sample matter $(kg/m^3)$. Generally, the weight of a sample could be calculated as follows:

$$m = \rho \cdot v \tag{3}$$

where $m$ is the weight of the seed (kg), $v$ is the volume of the sample $(m^3)$.

Within an ROI, the volume of each pixel intensity can be calculated by the product of a cross-section area of the pixel and the intensity at the pixel. Then, the total mass of seeds is the summed mass of all the pixels in the ROI (Equation (4)):

$$m = \sum (\rho \cdot \Delta s \cdot x_{ij}) \tag{4}$$

where $\Delta s$ is the cross-section physical area of each pixel in $m^2$; $x_{ij}$ represents the thickness at pixel $(i, j) (m)$, equivalent to the thickness of the seed in the pixel $(i, j)$ (Figure 2d). While the variations of total attenuation in the ROI mainly are due to the different seed weights, the weight of seeds in ROI can be described by combining Equations (1), (2), and (4), as follows:

$$m = \frac{\Delta s}{\gamma} \cdot \sum^{ROI} \ln\left(\frac{N_0}{N}\right) \tag{5}$$

where $N$ is the gray value of a pixel in the ROI, which is the intensity of an X-ray beam detected after traversing seed matter, and $N_0$ is the gray value of the same pixel that the intensity of an X-ray beam measured without any samples. Based on the two assumptions made for this study, $\frac{\Delta s}{\gamma}$ in Equation (5) is treated as a constant. The comprehensive attenuation index (CAI) is calculated by Equation (6):

$$CAI = \sum^{ROI} \ln\left(\frac{N_0}{N}\right) \tag{6}$$

where CAI is proposed in this study to sum up the X-ray attenuation caused by the peanut seeds in a pod. In this way, the CAI value can be calculated for each peanut to determine the degree of attenuation from an original X-ray image, and the slope coefficients, which include the information of the mass attenuation coefficient, can be fitted from the calibration set, since the linear relationship between the sample weight and CAI value is derived theoretically. Hence, linear regression models can be feasibly calibrated to transform the exponential attenuation information in X-ray images of peanut pods into the total amount of mass of peanut seeds.

### 2.5. Differential Evolution Segmentation Algorithm

Manual image segmentation is very time-consuming; thus, it needs to be automated [29]. The differential evolution (DE) algorithm has been widely used for numerical optimization problems because of its simplicity, robustness, and good convergence proper-

ties [30]. The Tsallis entropy function is commonly used for image thresholding [31]. In this study, the DE algorithm coupled with the Tsallis entropy function were applied to optimize the threshold determination to automatically obtain the ROI in a peanut seed image.

As shown in Figure 2a, the background, pod shell, and peanut seed are displayed as light, intermediate, and dark areas, respectively. The background area was the lightest because no attenuation occurred when the detector was directly exposed to the X-ray source. By contrast, the X-ray beam that passed through the peanut seed area was attenuated the most, showing a darker area in the image. The Tsallis entropy function was used to evaluate the segmentation results of dark, intermediate, and bright regions. The Tsallis entropy, $E_{Tsallis}$, was calculated by Equation (7):

$$
\begin{aligned}
E_{Tsallis} = &-(\sum_{i=0}^{\alpha}(\frac{p_i}{\sum_{i=0}^{\alpha} p_i}\cdot\ln(\frac{p_i}{\sum_{i=0}^{\alpha} p_i})) + \sum_{i=\alpha}^{\beta}(\frac{p_i}{\sum_{i=\alpha}^{\beta} p_i}\cdot\ln(\frac{p_i}{\sum_{i=\alpha}^{\beta} p_i})) \\
&+ \sum_{i=\beta}^{255}(\frac{p_i}{\sum_{i=\beta}^{255} p_i}\cdot\ln(\frac{p_i}{\sum_{i=\beta}^{255} p_i}))), (0 < \alpha < \beta < 255)
\end{aligned}
\tag{7}
$$

where $\alpha$ was the threshold for segmenting dark area from images, $\beta$ was the threshold for segmenting light area from images, and $p_i$ was the ratio of all pixels with a value of $i$ to the total pixels of an image. The target thresholds $\alpha$ and $\beta$ were optimized between the minimal pixel value of zero and the maximal pixel value of 255 in an 8-bit grayscale X-ray image. A higher entropy value indicated that pixels of peanut seed, pod shell, and background were accurately assigned to light, intermediate, and dark, respectively.

The DE algorithm was a population-based optimization technique for globally optimizing multi-modal functions, which was widely used for its simplicity, robustness, and good convergence properties [32]. In this study, the DE algorithm was used to optimize the thresholds $\alpha$ and $\beta$ by maximizing the designed metric $E_{Tsallis}$. In the DE algorithm, the thresholds $\alpha$ and $\beta$ were updated according to Equation (8) in each iteration:

$$
\alpha = \frac{\alpha_0 + \alpha_{optimal}}{2}, \beta = \frac{\beta_0 + \beta_{optimal}}{2}
\tag{8}
$$

where $\alpha$ was the updated value of $\alpha_0$, $\beta$ was the updated value of $\beta_0$, and $\alpha_{optimal}$ and $\beta_{optimal}$ were the optimal thresholds resulting from the optimal entropy in the previous iteration. To avoid local optimizations, the worst 25% of individuals were randomly regenerated in each iteration. During the initialization process, 100 individuals of the population and 50 iterations in total were set to ensure individual diversity and algorithm convergence. After all the iterations were completed, the thresholds resulting in the optimal entropy values were applied for segmentation.

### 2.6. Test of Significance

Equation (5) assumed that there was a strict linear relationship between the seed mass and the CAI value. Hence, the t-test and the F-test were applied to test the statistical significance of correlation coefficients by using experimental sample data, and an alpha level of 0.01 was selected to determine the significance of the statistic index. From another perspective, it was treated as a method to examine the correctness of the derived Equation (5).

### 2.7. Data Sets and Model Evaluation

Linear regression models were developed to predict peanut seed weights using the CAI calculated from the X-ray images. The performance of the developed models was evaluated using the determination coefficient ($R^2$) and the root mean square error (RMSE) for the calibration and the validation sets, respectively (Equations (9) and (10)):

$$
R^2 = 1 - \frac{\sum_{i=1}^{n}(y_i - \hat{y}_i)^2}{\sum_{i=1}^{n}(y_i - \overline{y})^2}
\tag{9}
$$

where $\hat{y}_i$ is the weight of $i_{th}$ sample calculated from model (g), $y_i$ is the measured weight of $i_{th}$ sample (g), $\bar{y}$ is the average weight of the samples in the calibration set (g), and $n$ is the total number of the samples in the calibration set.

$$\text{RMSEC(RMSEP)} = \sqrt{\frac{\sum_{i=1}^{n}(\hat{y}_i - y_i)^2}{n}} \tag{10}$$

where RMSEC is the root mean square error of the calibration set (g), and RMSEP is the root mean square error of predicting the validation set (g).

## 3. Results

### 3.1. Statistical Distribution of Seed Weights

Peanut pods within each cultivar were divided into a calibration set and a validation set, and the seed weights in the calibration set and validation set from each cultivar are summarized in Table 1. Boxplot diagrams were also drawn, as shown in Figure 3, to depict the distribution and range of seed weight in three cultivars.

**Table 1.** Number of pods and seed weights for the cultivars used in model calibration and verification.

| Cultivar | Calibration Set | | | | Validation Set | | | |
|---|---|---|---|---|---|---|---|---|
| | No. Pods | Range of Seed Weight (g/pod) | [1] SD (g) | Mean (g) | No. Pods | Range of Seed Weight (g) | [1] SD (g) | Mean (g) |
| Florida Fancy | 166 | 0.05~2.68 | 0.42 | 1.54 | 83 | 0.21~2.43 | 0.40 | 1.52 |
| OLé | 76 | 0.54~1.40 | 0.19 | 0.96 | 38 | 0.62~1.36 | 0.20 | 0.98 |
| Wynne | 100 | 0.20~3.76 | 0.69 | 1.90 | 50 | 0.32~3.07 | 0.61 | 2.04 |

Notes: [1] SD: standard deviation.

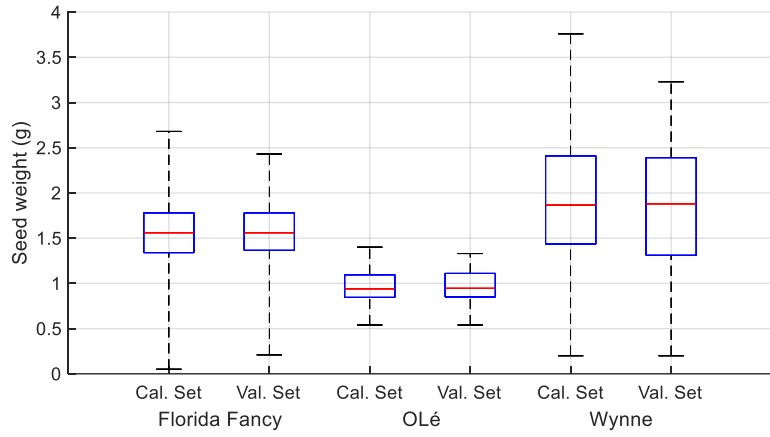

**Figure 3.** Boxplot diagrams of the distribution range of seed weight of cultivars Florida Fancy, OLé, and Wynne.

The distributions of seed weight of the three cultivars were different in their range of variability. The peanut pod of the Wynne cultivar was characterized by its large size, the seed weights of selected samples ranged from 0.20 g to 3.76 g/pod, with a mean of 1.90 g/pod in the calibration set. On the other hand, the peanut pod size of the OLé cultivar was much smaller, with a mean of 0.96 g/pod, ranging from 0.54 g to 1.4 g/pod in the calibration set. The peanut pod of Florida Fancy was the medium size of the three cultivars, with a range of 0.05~2.68 g/pod and a mean of 1.54 g/pod in the calibration set.

### 3.2. Correlations between CAI Values and Seed Weights

CAIs were calculated based on manual segmented ROI and DE segmented ROI, respectively. The correlation coefficients (r) between CAIs and peanut seed weights are presented in Table 2. The results showed that the linear correlation relationships were

significant at a level of 0.01 for all experimental data sets. All correlation coefficients were greater than 0.92, showing a strong positive linear correlation relationship between the CAI values and seed weights. Hence, linear regression models could be performed in further processes.

**Table 2.** Correlation coefficients between CAI values and peanut seed weights.

| Cultivar | Thresholding Methods | No. Pods Sampled | r | $t$-Statistic ($t_{(n-2)}$) | $p$-Value |
|---|---|---|---|---|---|
| Florida Fancy | [1] Manual | 249 | 0.9717 | 64.7078 | <0.001 |
| | [2] DE | 249 | 0.9626 | 55.8754 | <0.001 |
| OLé | [1] Manual | 114 | 0.9377 | 28.5606 | <0.001 |
| | [2] DE | 114 | 0.9253 | 25.8168 | <0.001 |
| Wynne | [1] Manual | 150 | 0.9822 | 63.6823 | <0.001 |
| | [2] DE | 150 | 0.9742 | 52.5599 | <0.001 |

Note: [1] manual: manual segmentation, [2] DE: differential evolution segmentation.

### 3.3. Regression Models for Estimating Peanut Seed Weights

The linear regression models between CAI values and seed weights, established after passing the $t$-statistic tests, and the parameters of the models are shown in Table 3. All the models performed high $R^2$ values and low RMSEC and RMSEP values, which passed the significance test of the F-statistic at a 99.9% confidence level. All $R^2$ values were greater than 0.85, which showed the linear relationship between the CAI values and the seed weights. The linear regression model with the manual segmentation method of the Wynne cultivar performed the best with an $R^2$ of 0.9672. The RESEP values from models of three cultivars ranged from 0.0756 g to 0.1463 g, in an equivalent range of 0.0617~0.1518 g among all RMSEC values. Meanwhile, it was observed that the models with the DE segmentation method performed as similarly well as those models with the manual segmentation method. In other words, the DE optimized thresholding method developed in this study could obtain as precise target ROIs as the manual segmentation method did. The combined models were calibrated by merging samples in three calibration sets from all cultivars. The $R^2$ values were 0.9652 and 0.9547 for combined models with manual and DE segmentation methods, respectively.

**Table 3.** The results of seed weight estimation models for the three peanut cultivars.

| Cultivar | Threshold Method | [1] Cal. Set | Model Coefficients | | RMSEC (g) | $R^2$ | [4] Val. Set | RMSEP (g) |
|---|---|---|---|---|---|---|---|---|
| | | | [2] $a$ | [3] $b$ | | | | |
| Florida Fancy | [5] Manual | 166 | $8.8338 \times 10^{-6}$ | −0.0046 | 0.0996 | 0.9440 | 83 | 0.0927 |
| | [6] DE | 166 | $8.6514 \times 10^{-6}$ | 0.0319 | 0.1132 | 0.9276 | 83 | 0.1084 |
| OLé | [5] Manual | 76 | $8.5294 \times 10^{-6}$ | 0.0739 | 0.0617 | 0.8932 | 38 | 0.0756 |
| | [6] DE | 76 | $8.2347 \times 10^{-6}$ | 0.1102 | 0.0661 | 0.8776 | 38 | 0.0845 |
| Wynne | [5] Manual | 100 | $8.6849 \times 10^{-6}$ | −0.0745 | 0.1247 | 0.9672 | 50 | 0.1248 |
| | [6] DE | 100 | $8.3274 \times 10^{-6}$ | −0.0058 | 0.1518 | 0.9514 | 50 | 0.1463 |
| [7] Combined | [5] Manual | 342 | $8.2836 \times 10^{-6}$ | 0.0714 | 0.1093 | 0.9652 | 171 | 0.1063 |
| | [6] DE | 342 | $8.0539 \times 10^{-6}$ | 0.1112 | 0.1246 | 0.9547 | 171 | 0.1207 |

Notes: [1] Cal. Set: number of peanut pods in the calibration set; [2] $a$: slope coefficient of the linear regression model; [3] $b$: intercept coefficient of the linear regression model; [4] Val. Set: number of peanut pods in the validation set; [5] manual: manual segmentation, [6] DE: differential evolution segmentation. [7] Combined: samples with a combination of the three varieties.

Figure 4 shows the prediction details for the calibration and validation sets in models with the DE segmentation method. The model for the Wynne cultivar performed the best, with an $R^2$ of 0.9514. The model for the OLé cultivar achieved a lower $R^2$ of 0.8775 than the other two cultivars. While merging the samples of three calibration sets from all cultivars into a larger calibration set, the combined model was slightly better than other models calibrated within samples of a single cultivar. The combined model with the DE segmentation method resulted in a higher $R^2$ value of 0.9547 and an RMSEP value of 0.1206 g, achieving an acceptable deviation level for practical applications.

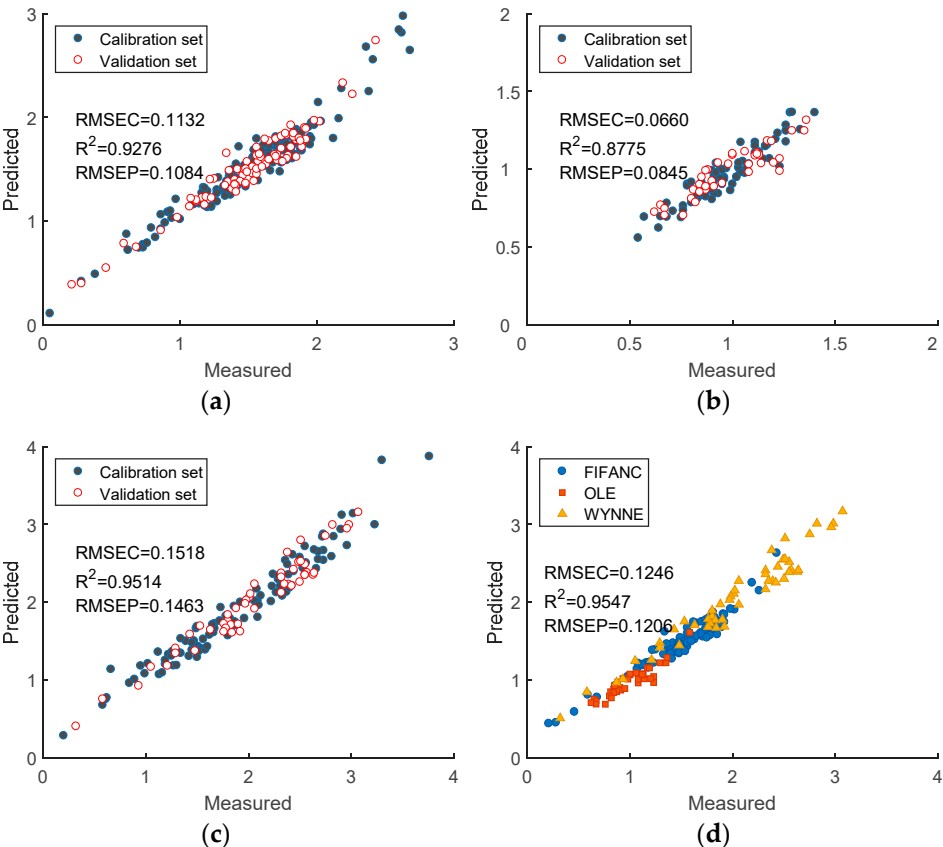

**Figure 4.** The prediction results of the models based on the DE segmentation algorithms for the calibration sets and the validation sets: (**a**) Florida Fancy; (**b**) OLé; (**c**) Wynne; (**d**) combined sample with all three cultivars.

## 4. Discussion

In this study, the soft X-ray imaging method was demonstrated to address the problem of estimating the seed weight of unshelled peanut pods. Three cultivars of peanut characterized in different pod sizes, namely Florida Fancy, OLé, and Wynne, were selected to represent peanut samples in a wide variety of pod sizes. At the same time, the differential evolution (DE) segmentation algorithm was designed as an ROI segmentation method to accommodate the changes in pod size. The results showed that the X-ray imaging method had significant potential advantages in measuring peanuts' seed weight without peeling the shell. This would contribute to the realization of automatic operation for peanut quality grading.

Based on the data in Table 2, the high correlation coefficients showed that there was a strong linear relationship of the proposed CAI indicator and peanut seed weight. By converting the exponential absorption information into a linear indicator, it provided a pathway to transform a two-dimensional (2D) pixel into a three-dimensional (3D) voxel for virtual volume computations, as previous studies reported [15,33]. It would also be more intuitive to show the calculation process and acquire the actual weight of peanut seeds.

According to the data in Table 3, the linear regression relationship was proved by the experimental data set at a 99.9% confidence level. The ROIs were extracted by a manual method, and DE method, the $R^2$ value of the model for the Wynne cultivar was the highest, while that of the model for the OLé cultivar was the lowest. This could be due to the differences in the size of the seeds and the number of samples included in the models. OLé had a smaller sample size and seed weight than the other two cultivars, and the physical size of seeds and pods of the Wynne cultivar were the largest among the three cultivars. This indicated that the relative error of the model for measuring large-size peanut samples would be reduced. On the other hand, the models with DE automatic segmentation method

performed as well as those models with manual segmentation method in each cultivar. The gaps between DE models and manual models might mainly result from the fact that the DE optimized thresholding method only considered the gray value of pixels, while the manual segmentation considered the morphological features visually of peanut pods and seeds beside the pixel values. Hence, the manual segmentation method could provide a more precise ROI for calculating the CAI value. However, the models based on the DE segmentation algorithm had advantages in the potential for automation.

The combined models were established to investigate the feasibility of developing a common model for estimating peanut samples of different cultivars. Subsequently, the combined model with the DE method obtained a slightly higher $R^2$ value of 0.9547 than other models with the DE method upon single cultivars, since a larger sample size resulted in a better fitting of the model coefficients. Meanwhile, it appeared that the seeds of the three cultivars had approximately the same mass attenuation characteristic. Hence, it could conservatively come to a conclusion that it could be feasible to establish a comprehensive method for estimating seed weights for different cultivated peanut genotypes in future applications.

While discussing the applications of the X-ray imaging technique, it has been usually compared with the X-ray computed tomography (CT) imaging technique [22,34,35]. Notoriously, the X-ray CT technique has been widely used for medical and non-medical applications since it has a higher resolution in 3D visualization analysis. However, the X-ray CT technique was still costly and time-consuming, and was not suitable for high-throughput batch detection of industry uses. On the contrary, the advantages of X-ray imaging were low cost, high speed, and the ability to handle a batch of samples at high throughput. In this study, it set an exposure time of 2 s to capture an X-ray image and an average 2 s to calculate the seed weight with a common desktop computer; this process can be shortened by increasing the peak voltage and/or current amplitude as well as improve the computing power. Domhoefer et al. [25] only spent 0.3 s on capturing an X-ray image of a peanut pod and also reported the time for prediction of features were approximately 1 s by the X-ray image transformation method and 6 s by convolutional neural network method with additional GPU hardware. Developing pipeline-type detection equipment and integrating the established model into the control system would become an effective way to realize high-throughput detection for the peanut industry by using an X-ray imaging technique. Meanwhile, further concerns regarding the potential issues that might arise should be addressed while implementing the X-ray imaging technique in the agricultural field. Firstly, the possible radiation hazard to the human body must be first considered because agricultural products are the main source of energy for human beings. The higher energy of X-ray beam would pose a higher radiation risk to living animals and plants, as well as humans who adopt these food as a source of energy. Therefore, it is necessary to verify that using X-ray radiation would not substantially alter the gene character of samples before applying the technique to a large scale. Secondly, the difference in attenuation properties between the target part and the background part in samples should be significant because the similarity in the attenuation properties would result in poor image contrast and pose a challenge in image segmentation. Thirdly, according to the service life of the X-ray imaging system, routine maintenance is required to ensure the accuracy and reliability of the detection results.

Based on the above concerns, more meaningful research could be carried out in future research. First of all, the effects on agricultural products such as crop seeds caused by X-ray beams with different radiation energies deserve to be studied. Furthermore, it is necessary to develop more robust and intelligent image analysis algorithms in the future, which could address more complex detection problems. Finally, it is also of great significance to study innovative technologies for producing X-ray light source generators and camera detectors in X-ray imaging systems, since improving the stability of the light source, as well as the sensitivity of the sensor, and reducing the energy consumption of the system are important factors to promote the popularization of X-ray imaging technique for industrial

applications. Overall, there are still many factors to be taken into account while developing an automated commercial system using a portable X-ray technique, including real-time consumption, high throughput batch processing capability, and economic costs.

## 5. Conclusions

This study successfully demonstrated a non-destructive method to estimate the peanut seed weight of intact peanut pods by using soft X-ray imaging and an automatic threshold segmentation algorithm. It is feasible to non-destructively measure the peanut seed weights by calculating the gray value of pixels on an X-ray image of unshelled pods. The differential evolution (DE) algorithm, designed to obtain the region of peanut seeds within the X-ray image, could automatically extract the target ROI as precisely as manually. The comprehensive attenuation index (CAI) value was calculated from the segmented ROIs. The high correlation coefficients of CAI values and actual seed weights showed a strong positive linear correlation relationship between these two variables. Furthermore, the high determination coefficients in experimental results verified that linear regression models could be used for predicting peanut seed weights with an unknown independent sample set. Overall, it demonstrated that the X-ray imaging technology, coupled with the differential evolution segmentation algorithm, can be used to estimate seed weights from intact peanut pods non-destructively and efficiently. More work is needed to fine-tune the developed models and invent equipment with additional experimental studies, promoting this technology to produce huge economic benefits in the peanut value chain.

**Author Contributions:** Conceptualization and methodology, N.W., R.S.B., G.Q., and Y.L.; data collection and analysis, Y.L. and G.Q.; original draft preparation, G.Q.; review and editing, N.W., R.S.B., and P.R.W.; project administration, N.W. All authors have read and agreed to the published version of the manuscript.

**Funding:** This research was funded by the USDA-NIFA Hatch Projects: OKL03169 and OKL03178 and also supported by the Natural Science Foundation of Guangdong Province (2022A1515010391); the Innovation Fund of Guangdong Academy of Agricultural Sciences (No. 202104); the National Natural Science Foundation of China (No. 42001256); and the Youth Training Program of Guangdong Academy of Agricultural Sciences (No. R2020QD-061).

**Conflicts of Interest:** The authors declare no conflict of interest.

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
