# Peer review of "Non-Destructive Method for Estimating Seed Weights from Intact Peanut Pods Using Soft X-ray Imaging"

_agronomy, doi:10.3390/agronomy13041127_

Round 1

Reviewer 1 Report

Please find below my observations:

1) has the soft X-ray imaging system been originally developed by the authors? If so, this should be better explained, even in comparison with other possibly existing systems

 2) is there the possibility to compare the performance of hard X-rays vs soft X-ray in performing this specific task? It is well known that the signal of soft X-rays is noisier and more difficult to reconstruct with standard techniques, as the one used in this context

3) uniform mass attenuation coefficient for peanut seeds was assumed: what is the value assumed? What is the impact of choosing an average value for it on the CAI? 

4) I am not surprised that an automatic segmentation could work fine as well, due to the relatively simply shape of the pod. Are there other manual phases left in the procedure?

5)   the Florida and Wynne  cultivars display the most relevant errors in the prediction of the seed weight, which can also represent an error > 60%. Is this relevant in grading the pods?

Author Response

Dear reviewer,

We appreciate you for the positive and constructive comments on our manuscript (ID: agronomy-2304342). We have studied all the comments carefully and have tried our best to revise the manuscript according to these comments. The responses and revisions have been completed item by item in response to the reviewers' comments.

We would like to express our great appreciation to you and the reviewers for new comments on our paper. Thanks for your time and looking forward to hearing from you.

Regards,

Ning Wang

Reviewer 2 Report

I have carefully reviewed your manuscript entitled "Non-destructive Method for Estimating Seed Weights from Intact Peanut Pods Using Soft X-ray Imaging" and would like to provide some feedback to improve the manuscript.

 Firstly, I suggest that the authors better discuss how the proposed technology can be implemented in the industry for high throughput performance.

 Secondly, the manuscript should more thoroughly address the limitations of the proposed technique. Readers would benefit from a detailed discussion of any potential issues that may arise when implementing the method in the field.

 Thirdly, I recommend that the authors include a more comprehensive discussion about future research topics. What are the next steps for the development of this technology, and what other applications could it have? This would help readers to understand the potential for future research and development in this area.

 In addition, I suggest that the authors include some extra recently published work in the manuscript which utilized AI tools and soft X-ray imaging to evaluate seed quality parameters, such as https://link.springer.com/article/10.1007/s11947-022-02939-5

Overall, I believe the manuscript is well-written, and the results are significant. With the suggested revisions, the manuscript will be even stronger and will contribute significantly to the field.

Author Response

Dear Reviewer,

We appreciate you for the positive and constructive comments on our manuscript (ID: agronomy-2304342). We have studied all the comments carefully and have tried our best to revise the manuscript according to these comments. The responses and revisions have been completed item by item in response to the reviewers' comments.

We would like to express our great appreciation to you and the reviewers for new comments on our paper. Thanks for your time and looking forward to hearing from you.

Regards,

Ning Wang
